# Computers as a Tool to Empower Students and Enhance Their Learning Experience: A Social Sciences Case Study

**David Antonio Buentello-Montoya**

Tecnologico de Monterrey, Escuela de Ingenieria y Ciencias, Zapopan 45201, Mexico; david.buentello@tec.mx

**Abstract:** Computers in mathematics education help foster abstract concepts and solve problems that are unsolvable by hand. Moreover, students whose major does not have a mathematical background often struggle with the topic and may require learning aid. Although extensive research has been conducted on the effect of computers and different software in learning, students' perception on computers to solve mathematical problems has scarcely been studied. In this work, a group of undergraduate social sciences students were given computers to learn mathematics and solve contextual real-life problems, with the aim of facilitating learning and providing empowerment. After the courses, the students were asked their perception of computers and mathematics to obtain descriptive results on their perception. Responses indicated that with computers, students felt learning and solving mathematics problems was easier (85% of the answers), they felt more confident about their skills (69%), and could think of new ways to solve problems (54%). Moreover, a text analysis was conducted using software to assess open-ended questions, and the results indicated that most answers were positive in nature. Additionally, the students were given the chance to rate the mathematics course using computers, and the course received a rating of 9.5/10, indicating the methodology was liked.

**Keywords:** technology-enhanced learning; computers in mathematics teaching; innovative learning experiences; educational innovation; programming in mathematics teaching; contextual mathematics

## 1. Introduction

A computer is an electronic device that can store and process data based on instructions given by a program or programmer [1]. Computers have diverse applications and can be found in many forms and not just as desktop or laptop computers, as devices such as telephones and televisions can be considered computers. Computers are fundamental and have revolutionized several aspects of human life, including education. In education, calculations, representations, and communication are common aspects reinforced with computer assistance; however, even gaming with a computer can be channeled into a pedagogical activity if the activity is designed properly [2]. At the same time, computers can be used as a learning aid, and the emergence of technologies such as Artificial Intelligence (AI) can serve to harness the endless amount of information available on the internet.

Calculations are the most common use for computers: from Computer Assisted Design (CAD) software to spreadsheets, calculations provide users with information that would be otherwise difficult to obtain and/or display in a readable manner. On the other hand, understanding mathematics and how it relates to different phenomena is the backbone of education. Therefore, calculations and representations (which result in simulations) are extremely useful for teaching and learning different topics such as science. For example, PhET interactive simulators provide teachers and students the opportunity to conduct and observe different experiments involving physics and chemistry (for example, orbits, ideal gases, and electromagnetism-related phenomena) using a web application [3]. However, and importantly, even if computers are capable of displaying plots, graphs

and other kinds of results, the results must be interpreted, as they will otherwise just be meaningless numbers.

Representations (also called visualizations) are visual multimedia representations of different objects that can help students understand different concepts. Representation covers visual aspects, from simple static images, scatter plots, and pie charts, to graphing mathematical functions using software such as Geogebra [4]. Moreover, novel reality-enhancing technologies such as 3D representation, as well as virtual, augmented, and mixed reality are examples of representations [5].

Communication in education not only refers to student−student and teacher−student interaction, but also includes accessing all kinds of communication resources available using a computer/mobile phone/tablet. E-books, recorded lectures, and short clips for Just-in-Time Teaching are examples of communication-based learning enablers [2]. Moreover, video websites, such as YouTube and TikTok, and Massive Open Online Courses (MOOCs) can be considered communication-based learning enablers for computer assisted instruction.

At the same time, it is known that computers are much better than humans for "algorithm solving" and have better capabilities for graphing; hence, they are tasked with doing most of our mathematical calculations and visualization tasks. However, humans are required to manipulate computers, give orders, and interpret the result (output), as otherwise the solutions will not make sense (for example, vector directions, negative area/volume values, and negative time values) [6].

On the other hand, the literature indicates that students may struggle with mathematics because they have a low level of self-efficacy (that is, they do not consider themselves proficient enough in mathematics) [6–8]. As stated by Conrad Wolfram [9], the way mathematics are taught must change to adapt to society's needs. Mathematics is often invisible to workers as people rely on "black box" approaches where the user receives an output from a simplified input [10]. Therefore, it is necessary for professionals to understand the uses and relationships between mathematics and computers; students should be taught how to complement computer capabilities, instead of trying to compete with them [6]. Moreover, social sciences students are often daunted by mathematics and may feel overwhelmed and lost in the abstract concepts; for this reason, besides computers, contextual mathematics activities (called Model Eliciting Activities (MEA)) have been incorporated in activities [11,12] to motivate students and improve their learning experience. As explained by Rubel and McCloskey [13], contextualized mathematics motivates students and encourages and supports learning. Not all assignments should be MEAs, however, as this could result in fading novelty and cognitive overload when trying to relate everything to mathematics. Whenever students worked with MEAs, prior to contextualization, every topic was explained abstractly. Mathematics education, together with computational education and critical thinking, can contribute to the formation of new-generation professionals, ready to tackle the challenges of volatility, uncertainty, complexity, and ambiguity (VUCA) [14]. Computers have been used to teach mathematics using different software and tools [15–18]. In contrast, in this research, different software was used so students could understand how computers as a whole can be useful for problem solving, particularly in the case of mathematics, and how they could boost their abilities in circumstances where students would otherwise struggle.

In this work, the effect of using a computer to enhance students' confidence when learning of mathematics and their mathematic problem-solving capabilities was analyzed descriptively. For this goal, students were taught how to manipulate computers to solve their mathematical problems using software, and to interpret the obtained results, and the opinions on the employed methodology were collected and assessed. Moreover, by using a computer to solve mathematical problems, students were empowered with a sense of achievement. A sense of achievement is a contributing factor to self-efficacy, which translates into engagement, motivation, and improved learning and performance [7,8,14]. Measuring a person's mathematics self-efficacy and anxiety is not a trivial task and requires extensive questioning and study [7,19]. Instead, this work measured and determined

whether students felt empowered by using a computer to study mathematics and solve otherwise complicated problems. Although not formally defined as a component of self-efficacy, a sense of empowerment can provide an idea of the effect of the computer in the learner's self-efficacy and potential learning outcomes.

To determine the impact of the experiment, a questionnaire to learn their opinions and thoughts was designed and employed. The answers from the questionnaire were used to answer the following research questions (RQ):

RQ1. Will using a computer to solve mathematical problems make mathematics learning feel easier?

RQ2. Does solving mathematical problems with a computer provide students with a sense of empowerment?

The structure of the work is as follows. Section 2 presents a brief literature review related to the use of computers in teaching mathematics as well as model-eliciting activities. Section 3 describes the methodology followed in the work. Firstly, the participants in the study and the course are described, followed by examples of how computers are used to improve fostering mathematical concepts and solving problems. Finally, the section presents the methodology used to measure the impact of the study and provides details on the questionnaire the students answered. Moreover, as some of the questions had qualitative answers, the methodology used to analyze these responses (text analysis) is also explained in Section 3. Section 4 presents the results from the findings and elaborates on the responses to the questionnaire, as well as the qualitative answers. Section 5 presents a discussion on the results and the answers to the research questions. Finally, a summary and the conclusions obtained from the study are conferred in Section 6, which serves as the closure to the manuscript.

## 2. Literature Review

The following subsections present a brief review of the literature on the use of computers in mathematics teaching, as well as contextual teaching−learning activities.

### 2.1. Computers in Mathematics Teaching

Although computers have been used to teach mathematics for a relatively long time, their use is mostly related to employing software and focuses on improving student performance [20–24].

Bernard et al. [25] used a combination of computers and contextual mathematics to improve students' understanding of mathematic concepts such as fractions by raising their confidence. To improve their confidence, students used Visual Basic Learning Media Applications in Excel. When comparing the students' ability to perform mathematic operations with fractions before and after the experiments, a significant improvement was seen, indicating that the computers improved the students' performance.

Li and Ma [26] examined the effects of using computers to teach mathematics in K-12 level. The researchers found that while computers aided mathematics learning, several aspects such as the teacher and the method of implementation could affect the learning outcome.

Taleb et al. [27] assessed whether mobile learning (also known as m-learning) has an impact on students and teachers when learning and teaching mathematics. The authors conducted their experiments in secondary schools and determined that both the students and teachers had positive perceptions regarding the use of mobile learning, and that the students felt particularly engaged by the methodology.

Zaldívar-Colado et al. [20] developed and used computer software to aid elementary school students in understanding arithmetic and basic algebra, and the researchers found that although most students reported that it was easier to learn mathematics using software, it was also easy to become distracted due to having access to different activities in a computer.

Susanti et al. [28] studied whether computers impacted the performance of students in mathematics courses at different levels (middle and high school) by comparing an experimental and a control group. The researchers found that middle-school students exhibited differences in performance, but no differences in higher-thinking skills were found.

Brezavscek et al. [29] conducted an analysis to relate different aspects in the learning of mathematics in secondary school and found that the adequate use of technology contributed to successful learning, even if the impact was small when compared with other factors such as learning willingness.

Istikomah and Wahyuni [30] studied anxiety in students using computers to learn mathematics and found that students felt more anxious about the software usage than about the mathematics, due to unpreparedness and uncertainty regarding the software being used.

Appavooa [31] used a technology-enhanced-learning approach to teach elementary school students the concept of fractions using a tablet. Among other results, the author found that students liked the methodology and were eager to use technology in additional courses, and that students that were already doing well in mathematics further improved their performance with the inclusion of tablets in their studies.

### 2.2. Contextual Mathematic Activities

Model eliciting activities were first documented by Scott Chamberlin [32]; they are used to establish a contextual problem to aid in learning, where students develop models applied to real-world situations. Among other characteristics, model-eliciting activities are based on the final solution to problems and can be directed towards the development of a particular piece of knowledge or skill in students.

Diefes-Dux et al. [33] used contextual mathematics activities in an undergraduate engineering course to increase the interest of women in engineering. As an example, the authors presented students with a hands-on experience of a nanotechnology practice.

In a different experiment undertaken at the University of Pittsburg [34], model eliciting activities encompassing a wide array of disciplines (such as industrial engineering) were used to immerse students in a client-driven learning experience; the researchers found that an important success-defining aspect of model-eliciting-activities-based teaching lies in the companionship provided by the facilitator.

Root et al. [35] used contextual mathematics to teach arithmetic; particularly, the researchers associated arithmetic with finance concepts, such as percentages and discounts. The researchers found that although more evidence is needed, contextualization did provide aid in teaching concepts by providing students with a sense of importance regaridng mathematics.

Syamsuddin and Istiyono [36] used a contextual approach to teach mathematics to junior high-school students. To measure the impact of the methodology, the researchers used a 12-question survey, and the results indicated that overall, students liked the employed methodology. Moreover, although not formally measured, the involvement of the students in learning activities showed that students were engaged; hence, the contextual approach helped engage students.

Mentari and Syarifuddin [37] used contextual teaching and learning with junior high-school students from Indonesia, and measured cognitive engagement, surface strategy, deep strategy, and reliance, which, as a group, indicated whether students paid significant effort to understanding lessons and whether students were willing to engage and become involved in learning activities. The results obtained by the researchers indicate that students were engaged by contextual activities but needed to be given room for creative thinking and answering in order to make the most of the methodology.

Lestari et al. [38] implemented a contextual teaching strategy in an elementary school with the aim of improving the students' critical thinking. For their implementation, the researchers used a comic book, where in the comic book, mathematical concepts were taught. The results indicated that students found the comic books attractive, and this attraction

turned into increased interest and willingness to learn in students, further translated into improved critical thinking.

### 2.3. Findings from Literature Review

All in all, from the literature review, the contextual teaching and learning methodology has been used since the 1970s, following different approaches and at different academic levels. At the same time, computers have been used in education extensively. However, most times, the methodology is used to improve learning, and it has been seldom used to make learners feel "more capable, adept, or empowered". Moreover, the effect contextual mathematics and computers on learning and students' perception has not been tested together. Based on this finding, this research serves to bridge the gap in knowledge by exposing whether empowerment and a sense of accomplishment can allow students' feeling of self-efficacy regarding mathematics increase.

## 3. Methodology

### 3.1. Description of the Participants, Courses, and Lectures

A group of social sciences students (N = 67) undertook mathematics courses in a private university throughout a one and a half-year period (three semesters); the students were aged between 17 and 21 years (during their first semester), and the female to male ratio was 4:1 (that is, there were four females for each male in the group). All of the students agreed to participate in the study, which started during their first semester and would end with their third semester. Finally, the same teacher instructed the mathematics courses during the three semesters.

The mathematics courses where the study took place were differential calculus (first year, first semester), integral calculus (first year, second semester), and multivariable calculus (second year, third semester). Although the courses were taught to social sciences students, the courses' syllabus was akin to that of engineering calculus courses (that is, included concepts such as limits, functions, derivatives, and vectors). During some lectures, after explaining the concept and theoretical background, the teacher used the computer to provide teaching−learning support to varying extents. Moreover, as some assignments required students to use a particular software to reflect and/or solve, the teacher provided tutorials, guidance, and resources (in the form of e-books and videos) on software usage. A breakdown of the topics, the aspect reinforced by including a computer, and the employed software is found in Table 1.

**Table 1.** Description of the computer-assisted instruction methodology employed to teach different topics.

| Calculus Course | Topic | Methodology | |
| --- | --- | --- | --- |
| Differential | Functions | Visualization | Geogebra, Excel |
| Differential | Derivatives (geometric interpretation) | Visualization | Geogebra |
| Differential | Optimization | Visualization | Geogebra |
| Integral | Antiderivatives | Calculation, visualization | Geogebra |
| Integral | Riemann sum | Calculation | Excel |
| Multivariable | Functions | Calculation, visualization | Geogebra |
| Multivariable | Derivatives and higher order derivatives | Calculation | Python |
| Multivariable | Lagrange multipliers | Calculation | Python |
| Multivariable | Kuhn−Tucker conditions | Calculation | Python |
| Multivariable | Iterated integrals | Calculation | Python |

The easiest to manipulate software (Geogebra, version 6.0) was used first and Python (the most complicated computational instrument) was only used in multivariable calculus to solve the most complex of problems that could not be solved with Excel nor Geogebra. As Python is a programming language that requires skills not conventionally found in

social sciences students, it was avoided during the differential and integral calculus courses so as to prevent cognitive overload and was only employed when revisiting topics that had similar contents to those previously studied [39]. The SymPy library for Python was employed for symbolic mathematics [40].

### 3.2. Examples of Contextual Mathematics

Some of the model eliciting activities used, along with the corresponding topics, are explained as follows:

- Functions: The students were asked the amount of money they thought would be convenient for retirement and had to make a plan for a monthly investment to make sure their savings were enough to meet their projections. The students modelled compound interest using polynomial functions, powers, and Euler's number. For their calculations, the students used Microsoft Excel. After their calculations, the students were asked to reflect on whether the calculations fulfilled their expectations, and students were tasked with providing a conclusion on whether their feelings about retirement changed.

- Optimization: For optimization, students were given statistics of how much a company spent on advertising and the associated revenue. Using Excel, the teacher presented the students a polynomial function to represent the revenue as a function of advertising expenses. Afterwards, the students used Geogebra to identify (graphically) the optimal advertising expenditures to maximize the revenue (at the point where the slope was zero). After identifying the optimal value graphically, students used derivatives to find the maximum revenue and its associated advertising expense and compared their calculations with the graph.

- Multivariable optimization: The students were tasked (in groups) with finding the optimal aspects where families had to spend their income to obtain the best value. The students conducted research on the living expenses of families from different socioeconomic levels and proposed a mathematical model as a function of the money spent on meals, rent, leisure, and transport. In the end, students had to optimize using Lagrange multipliers and Kuhn−Tucker conditions. As the problem consisted of a set of five equations with five unknowns in non-linear functions, which is not a trivial task, the students had to use Python to solve for the unknowns. Students were taught some of the basic functions of the SymPy (version 1.10) and NumPy (version 1.22) libraries [40,41], but the students had to solve the problem and find the optimal values by themselves. The students were not tasked with any particular way of developing their code and ended up developing different codes. The results obtained by each group of students were different, and each team had to interpret and reflect on the obtained results.

### 3.3. Impact Measurement

A questionnaire was used to measure the impact of the experiment. Although the questionnaire was not as exhaustive or extensive as other studies, it was designed based on the Program for International Student Assessment (PISA) mathematics anxiety test and the Pierce mathematics attitude scale [42,43], with similitudes to other reported scales and questionnaires, such as the Mathematics Anxiety Scale for Students test [42] and the Math Anxiety Questionnaire [43]. For the answers, a five-point Likert scale questionnaire was used, where 1 corresponded to "completely disagree" and 5 corresponded to "completely agree". The questions were as follows:

1. In a mathematics course, I can easily obtain the highest grade.
2. I can solve mathematics problems and homework without problems.
3. I understand how mathematical thinking will improve my graduation profile.
4. Every professional should know about mathematics and computers.
5. Using a computer to solve mathematical problems makes me feel secure about the answer.

6.  Technology has helped me better understand mathematics.
7.  Technology makes solving mathematical problems easier.
8.  Computers and mathematics have made me think of new ways to solve problems.

Questions 1–3 provide information on the students' perception of mathematics. Question 4 informs how students felt about computers. Questions 5 to 8 show whether there were changes in the perception of mathematics when computers were used, influenced by the methodology used throughout the course. Additionally, three open-ended questions were included. The first two open-ended questions asked something specific for a qualitative answer, while the last question was an empty field where students could add additional thoughts and comments; not all students answered the open-ended questions, however:

1.  If you agree (or disagree), explain why you think computers are essential (or not) in teaching and learning.
2.  If you agree, explain why the course changed your opinion about the overall use of computers.

Finally, the students gave their overall opinion on the courses and lectures, on a scale of 0 to 10.

*3.4. Text Analysis*

Since some of the answers to the questionnaire (open-ended questions 1–3) were qualitative in nature, the responses needed to be analyzed using text analysis tools. In order to study the most commonly used words and writing patterns and to conduct a sentiment analysis, Voyant Tools 2.2 [44] and the VADER 3.3 Python library [45] were used. Voyant Tools was used to generate Word Clouds and Word Link diagrams, and to obtain a positive to negative word ratio, while VADER was used to obtain a compound score, which indicates the overall lexicon rating of a sentence (between $-1$ as the most negative and 1, as the post positive) [45]. Importantly, word clouds provide an indication of the frequency of words, but not their relevance in a particular context; hence, a word link diagram can be useful.

## 4. Results

*4.1. Answers to the Likert-Scale Questionnaire*

The answers to the questions, in percentages, are portrayed in Table 2. The answers to questions 1 and 2 indicate that students have diverse perceptions of mathematics, as well as different levels of mathematics anxiety and self-efficacy; the evaluation of such constructs is outside of the scope of this work, however. On the other hand, the answers to Q2 show that most students neither completely agreed nor disagreed that they could solve mathematics problems. Arguably, this indicates students were uncertain about their actual mathematics skills.

**Table 2.** Questions asked in the survey, and answers given by students, in percentages; 1 stands for "completely disagree", while 5 stands for "completely agree".

| Question | Percentage Answers | | | | |
|---|---|---|---|---|---|
| | 1 | 2 | 3 | 4 | 5 |
| 1 | 23.08 | 15.38 | 23.08 | 15.38 | 23.08 |
| 2 | 15.38 | 0 | 30.77 | 38.0 | 15.38 |
| 3 | 0 | 0 | 8 | 23.08 | 69.23 |
| 4 | 0 | 0 | 8 | 7.69 | 84.62 |
| 5 | 0 | 0 | 0 | 30.77 | 69.23 |
| 6 | 0 | 0 | 15.38 | 30.77 | 53.85 |
| 7 | 0 | 0 | 0 | 7.69 | 92.31 |
| 8 | 0 | 0 | 0 | 23.08 | 76.92 |

The answers to Q3 and Q4 indicate that after using contextual mathematics and computers, students understood the importance of mathematics in their profile, and why it was included in the career curriculum (77%), and 92% completely agreed that every career should include the study of mathematics.

The answers to Q5 and Q6 showed that most students completely agreed computers made them feel more secure when solving mathematics problems (69%) and that computers made mathematics easier to understand (69%). Finally, the answers to Q7 and Q8 indicated that the majority of the students completely agree that computers made solving mathematical problems easier (85%) and that now they could think of new ways of solving problems (54%). These results were related to the sense of accomplishment and empowerment the students felt after they solved apparently complicated problems. As explained by Bandura [8], the sense of accomplishment led to an increase in self-efficacy; hence, this should have translated into a better overall performance.

Regarding the overall opinion on the course and the lectures, the students rated the course with a 9.5/10, with a standard deviation of 0.76, indicating satisfaction with the course and lectures. All in all, the received comments were positive and reflected a good attitude towards using computers in mathematics, and showed students with consciousness of the uses of a computer with mathematics felt more confident. Future work will monitor the performance of students before and after they felt this accomplishment.

### 4.2. Answers to Open-Ended Questions

The three open-ended questions (OEQs) presented qualitative results; hence, they required a different kind analysis. The answers were read, and answers that did not provide information (that is, answers that only included comments not related to the questions) were left out of the analysis. In the end, 56 answers were collected for the questions. To prepare the text, the answers were proofread to remove spelling mistakes. Afterwards, word clouds and sentiment analysis were conducted; the words "computer" and "computers" were left out of the word cloud (but were included in the word link diagram, for the sake of readability) as they did not provide much information due to the nature of the questions.

With regards to OEQ1, a word cloud and word link depicting the most common words and concepts used by the students is shown in Figure 1. The answers were diverse in nature. It can be seen in Figure 1a that the most used words were "solve" (27 times), followed by "problems" (25), "mathematics" (25), and "easier" (24). As portrayed in Figure 1b, the word "computer" was linked to mathematics, understand, simpler, visualize, thinking, and requirement; the word "problems" was linked to understand; and the word "solve" was linked to complicated, mathematical, operations, understand, simpler, and visualize. From the word link analysis, it can be inferred that the three mentioned main words were frequent because they served as "keywords" for the impact of computers in learning, in agreement with the answers provided by the students. Examples of the answers to OEQ1 were "visualization software gives sense to otherwise abstract concepts", "if we know how to use computers for calculations, we can spend more time understanding principles instead of trying to solve complicated operations", "computers help you solve problems that would be otherwise impossible", and how "mathematics is not as complicated as it seems if you have the appropriate tools". Additional results from the text analysis can be found in Table 3. The sentiment analysis indicated that most of the responses to OEQ1 were positive in nature, although the prevalence of words such as "problems" affect the positive-to-negative word ratio. Still, a positive-to-negative ratio larger than one, and a positive compound score indicates that overall, the answers could be thought of as positive in nature.

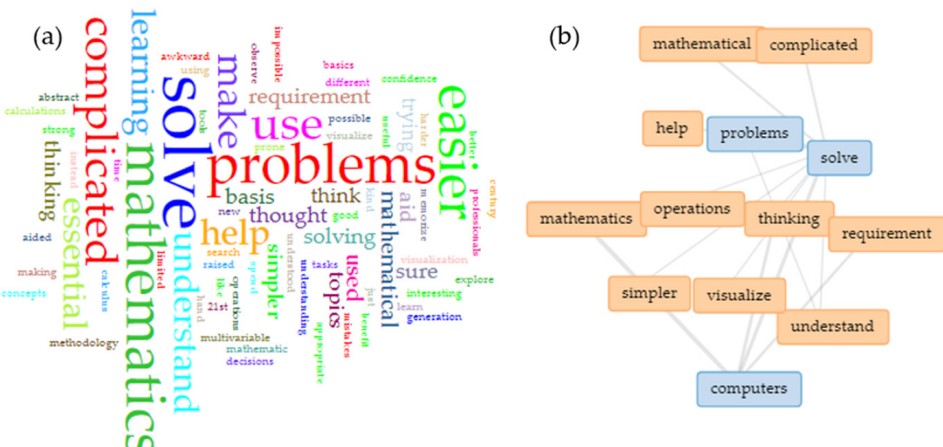

**Figure 1.** (**a**) Word cloud portraying the most common words, and (**b**) word link diagram, obtained from the students in the answers to open-ended question 1, "If you agree (or disagree), explain why you think computers are essential (or not) in teaching and learning.".

**Table 3.** Results from the sentiment analysis conducted using Voyant Tools and VADER.

| Open-Ended Question Number | Positive-to-Negative Word Ratio | Compound Score | |
|---|---|---|---|
| | | Mean | Standard Deviation |
| 1 | 1.571 | 0.242 | 0.360 |
| 2 | 3.667 | 0.377 | 0.259 |
| 3 | 3.5 | 0.234 | 0.254 |

With regards to OEQ2, "If you agree, explain why the course changed your opinion about the overall use of computers", the most common words are shown in Figure 2. The most common word was followed by "useful" (43 times), "might" and "know" (both found 31 times), and "software" (28 times). The word link diagram shows that "computers" was linked to wildly, use/used, mathematical/mathematics; the word "wildly" was linked to future, mathematics, and useful; the word "know" was linked to example, analysis, useful, and course; and the word "course" was linked to thought, analysis, opinion, and mathematical. Example answers included "I really learned like never before. Computers should be used in every course", "Computers can be wildly useful if you have the right knowledge. For example, there are unlimited possibilities for making graphics and visualization", and "computers should be used in every class, in every level, so we are more acquainted with them and don't require extra lectures". The sentiment analysis indicated that most of the words used by the students had a positive connotation, and the positive-to-negative word ratio was much larger when compared with question 1. This could be related to the nature of the question, as OEQ1 involved overcoming hardships, while OEQ2 was about a change in mindset.

Finally, with regards to the open-ended personal comments (where the word cloud and word links are found in Figure 3a,b), the answers were diverse. Some students added no comments at all, but some others mentioned how "visualization software gives sense to otherwise abstract concepts", "if they knew computers were that useful for problem solving, they might have spent more time learning how to use them", "computers help you solve problems that would be otherwise impossible", and how "mathematics is not as complicated as it seems if you have the appropriate tools". The most common words were useful (22 times), understand (12 times), software (12 times), future (12 times), and taught (11 times). The word "understand" was found to be linked to taught, thinking, persons, related, notes, education, and gates; the word "taught" was connected to way, education, gates, and essential; and the word "useful" was linked to gadget, taking, skill, education, thought, notes, thinking, know, and developed. The most common word (useful) was found as a nexus to words of different natures such as nouns (gadget) and verbs (to know).

The compound scores and mean results from the sentiment analysis ere similar to OEQ2, and further clarified that the responses to the open-ended questions were positive in nature.

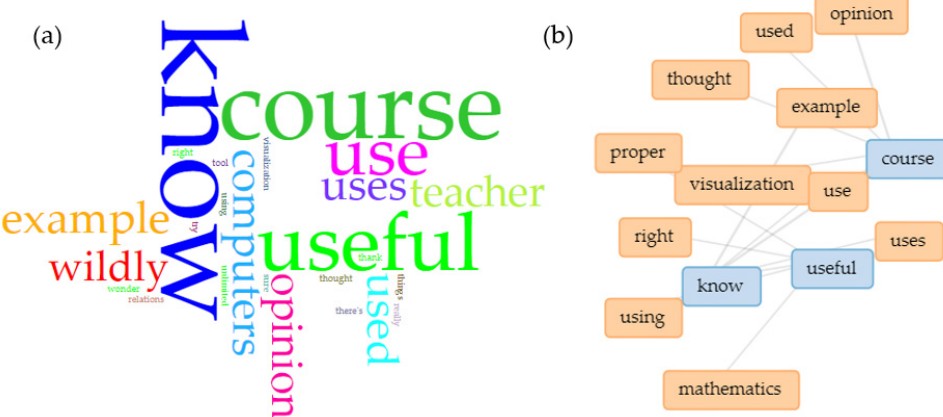

**Figure 2.** (**a**) Word cloud portraying the most common words, and (**b**) the word link diagram, obtained from the students in the answers to open-ended question 2, "If you agree, explain why the course changed your opinion about the overall use of computers".

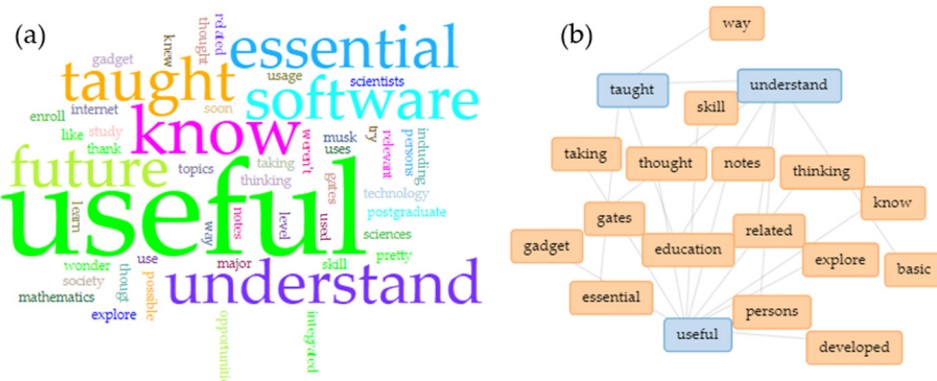

**Figure 3.** (**a**) Word cloud portraying the most common words and (**b**) word link diagram, obtained from the students in the answers to open-ended question 3.

### 4.3. Overall Opinion on Course

Finally, regarding the overall opinion on the course and the lectures, the students rated the course with a mean value of 9.5 (out of 10), with a standard deviation of 0.76. The low standard deviation indicates almost universal satisfaction, and praise to the course. All in all, the received comments were positive and reflected a good attitude towards using computers in mathematics, and showed that students with consciousness of the uses of a computer with mathematics felt more confident. Future work will monitor the performance of students looking for changes after the "feeling of empowerment" and will extend the use of computers to other disciplines such as chemical engineering.

### 5. Discussion

The results from the questionnaire indicated an overall positive response to using computers to assist in learning mathematics and solving mathematical problems. Q1 and Q2 had the most diverse responses. This was important, as there was a relationship between the level of mathematics anxiety, self-efficacy, and students' performance [7,46]. This can, in turn, affect how students feel about using computers to learn mathematics and solve problems. Although the performance of the students using computers for mathematics was not compared with a control group, the responses to questions 5, 6, and 7 indicated that as a whole, when using a computer, students found mathematics easier to understand,

mathematic problems easier to solve, and felt more secure when working with mathematics; hence, it could be concluded that computers made mathematics easier to learn, which could possibly translate into better grades when compared with students following a traditional lecture methodology. The found results are in agreement with those of Chen [47], who used Augmented Reality to improve the students' experience, make learning more fun and, finally, decreasing the students' mathematics anxiety. This was further supported with the analysis from the open-ended questions, where the responses were positive in nature. This answers RQ1, "Does using a computer to solve mathematical problems makes learning easier?": yes, using a computer eases the mathematics learning process, as even if the impact in student performance was not measured, there was a consensus in the students answers to questions related to ease of learning. On the other hand, the answers to questions 3–8 were all positive, and not even a single student answered 1–3 (from completely disagree to neither agree nor disagree) to the statements "I understand how mathematical thinking will improve my graduation profile", "Every professional should know about mathematics and computers", and "Using a computer to solve mathematical problems makes me feel secure about the answer", indicating students considered computers important in education. Moreover, the answers to questions 6–8 indicated students liked using computers and might have a new conception of the implementation teaching of computers in mathematics education. Furthermore, answers to the open-ended questions such as "I might study more about computers and software, now that I know how useful they are" point to the conclusion that students felt more secure and empowered when using a computer, answering RQ2, "Does solving mathematical problems with a computer give students a sense of empowerment?": yes, using a computer to solve problems gives a sense of empowerment and self-fulfillment, and may impact the students' self-efficacy. Moreover, based on the students' comments, using a computer and realizing the capabilities given opened up new opportunities and ideas, and might promote the development of transversal skills. As reported by Zay and Kurniasih [46], the students' computer self-efficacy was related to the students' mathematics self-efficacy; hence, if students feel confident about their computer skills, this will translate into confidence in mathematics skills. This presents a potential opportunity, as adequately digitally nurtured students could use computers as a gateway to mathematical thinking and towards the development of mathematical applications, regardless of their area of study. Additionally, the answers to the open-ended questions were in agreement with the findings reported by Appavooa [31], who interviewed students and found they were eager to use computers in additional topics, as they did not know they were as useful, and that learning with computers was much easier.

Authors such as Ochkov and Bogomolova [48] and Wolfram [9] have made mention about how students must be taught to manipulate mathematics and computers to achieve their goals, and mathematics and computers must be a tool and not an obstacle in fulfilling said goals; hence, different aspects of the students' mathematical skills and digital literacy must be measured. Future work will compare the performance of a group using computers with that of a group using a traditional methodology, to determine whether there is a change in grades (performance) and not just in attitude. Moreover, a larger analysis, where more students from different disciplines are involved, will be conducted; said analysis will also elaborate on the effect of using a computer on the anxiety, self-efficacy, and empowerment of students.

## 6. Conclusions and Future Work

A group of undergraduate social sciences students undertook mathematics courses for three semesters and used computers to assist the learning process and to solve mathematical problems. During the three semesters, the covered topics were differential, integral, and multivariable calculus. Depending on the studied topic, different software (Geogebra, Excel, Python) was employed during teaching and problem solving, and the students were asked their impressions on using a computer for mathematics. From the study, the following was concluded:

- Students felt comfortable using a computer to solve mathematical problems, and felt mathematics was easier to understand with the aid of computers.
- Students using a computer to solve mathematics better understood the relationship between both, their importance in their curriculum, and had more ideas regarding problem-solving opportunities.
- Text analysis results indicated that the students' perceptions of the uses of a computer in mathematics learning were positive.
- A computer can empower otherwise anxious students to try and solve problems.

As understanding and measuring engagement and self-efficacy is a complicated matter, there is definite room for improvement and for subsequent studies. Future work will include a comparison of the performance between a group using computers and a group following a traditional methodology, and will include an analysis on the students' anxiety and self-efficacy following standardized methodologies. Additional future work includes the use of computers to facilitate learning in other disciplines.

**Funding:** This research received no external funding. The APC was funded by Writing Lab, Institute for the Future of Education, Tecnologico de Monterrey, Mexico.

**Informed Consent Statement:** Informed consent was obtained from all subjects involved in the study.

**Acknowledgments:** The authors would like to acknowledge the financial support of Writing Lab, Institute for the Future of Education, Tecnologico de Monterrey, Mexico, in the production of this work.

**Conflicts of Interest:** The author declares no conflict of interest.

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
