# Peer review of "Computers as a Tool to Empower Students and Enhance Their Learning Experience: A Social Sciences Case Study"

_education, doi:10.3390/educsci13090886_

Round 1
Reviewer 1 Report
The study is very interesting, as it provides a positive perspective on the application of new technologies in education, within the current great controversy over whether ICTs are so beneficial for learning or not.
A complete bibliographical review and an adequate specification of the objective pursued are carried out.
The methodological part is correct, but I consider it necessary to specify more data on the validation process of the questionnaire, content analysis and reliability.
The presentation of the results is adequate and there is a good discussion of them.
Likewise, the conclusions are pertinent and concrete.
Author Response
Dear reviewer,
We appreciate your valuable comments which will surely improve the submitted manuscript. We have worked in addressing the following statement:
“The methodological part is correct, but I consider it necessary to specify more data on the validation process of the questionnaire, content analysis and reliability.”
Conducting a validation for a questionnaire is a complicated task, and we will be sure to conduct the appropriate tests and analysis for a future work. For the time being, we conducted additional literature review and included in the manuscript that our questionnaire has similitudes with other tests used to measure similar aspects (page 7, line 290-295):
“Although the questionnaire was not as exhaustive nor extensive as other studies, it was designed based on the Program for International Student Assessment (PISA) mathematics anxiety test and the Pierce mathematics attitude scale [42], [43], with similitudes to other reported scales and questionnaires such as the Mathematics Anxiety Scale for Students test [44] and the Math Anxiety Questionnaire [45].”
Kind regards.
Reviewer 2 Report
Feedback and suggestions can be seen in the manuscript

Author Response
Dear reviewer,
We thank you for the contributions to the manuscript. Please find attached a document where the changes to the manuscript are described in details.
Best regards.

Round 2
Reviewer 2 Report
improvements have been made based on suggestions